# Glioblastoma stem cells show transcriptionally correlated spatial organization
Shamini Ayyadhury [1,2] ✉, Patty Sachamitr[3,4], Michelle M. Kushida[3], Nicole I. Park[3], Fiona J. Coutinho[3], Owen Whitley[1], Panagiotis Prinos [4], Cheryl H. Arrowsmith [2,4,5], Peter B. Dirks [3,6,7,8,9], Trevor J. Pugh [2,5,10] & Gary D. Bader [1,2,6,11,12,13] ✉

Glioblastoma (GBM) is an aggressive brain cancer with a poor survival rate. Despite hundreds of clinical trials, there is no effective targeted therapy. Glioblastoma stem cells (GSCs) are an important GBM model system. In culture, these cells form spatial structures that share morphological aspects with their source tumors. We collected 17,000 phase-contrast images of 15 patient-derived GSC lines growing to confluence. We find that GSCs grow in characteristic multicellular patterns depending on their transcriptional state. Interpretable computer vision algorithms identified specific image features that predict transcriptional state across multiple cell confluency levels. This relationship will be useful in developing GSC screens where image features can be used to identify how GSC biology changes in response to perturbations simply by imaging cultured cells on plates.

Glioblastoma (GBM) is a brain cancer with poor overall survival despite extensive research and therapeutic testing[1,2]. Standard of care has remained unchanged, and newer therapeutic interventions have repeatedly failed in clinical trials. Heterogeneity, both across individuals and within tumors, including at the tumor-initiating glioblastoma stem cells (GSCs) level[3–6], may explain why it has been challenging to target GBM. A landmark TCGA study first identified that this heterogeneity has similarities to normal brain development and can be categorized into different lineage-representing groups of cells[7]. Single-cell gene expression experiments have since shown that GBM and GSCs maintain neurodevelopmental and injury-response hierarchies[3–5,8,9]. Brain developmental and differentiated cell-types and cell-states, such as the progenitor, astrocytic, oligodendrocytic and neuronal-like cell types, are found in all GBM tissues, often with a mesenchymal lineage that is transcriptionally similar to the astrocytic lineage[3,5]. Subsequently, studies showed that injury-response signatures are found enriched in GSCs, supporting the role of the reactivation of wound healing mechanisms in progenitors[4,9]. Ideally, improvements to GBM experimental model screening technology could increase the chances and speed of finding new therapies[10–12].

Current high-throughput therapeutic screening technologies are imaging-based, using either fluorescence or brightfield microscopy. Organoids and two-dimensional (2D) cell culture models are widely used in high-throughput screening experiments. These screening experiments capture aspects of tissue-level cellular organization and are valuable in therapy development and preclinical testing[13]. An advantage of 2D cell culture systems is that they can be easily imaged, capturing interesting aspects of cellular organization, including cell shape, spatial relationships between cells, and general properties of the patterns formed by cells on the plate. They also ensure that every cell can be similarly influenced by media conditions. Traditional image-based therapeutic screening methods applied to cellular culture models generally extract few and relatively general phenotypes, such as growth rate or expression of select protein markers, from measured images[14,15]. Circumventing these limitations, recent image analysis advances support extracting rich, multidimensional information from single-cell images[16–18]. However, most methods do not consider the organization of cells into multicellular structures and how these patterns relate to the human tissues modeled by 2D cultures.

[1]The Donnelly Centre, University of Toronto, Toronto, Canada. [2]Princess Margaret Cancer Centre, University Health Network, Toronto, Canada. [3]The Arthur and Sonia Labatt Brain Tumor Research Centre, The Hospital for Sick Children, Toronto, Canada. [4]Structural Genomics Consortium, University of Toronto, Toronto, Canada. [5]Department of Medical Biophysics, University of Toronto, Toronto, Canada. [6]Department of Molecular Genetics, University of Toronto, Toronto, Canada. [7]Division of Neurosurgery, Department of Surgery, University of Toronto, Toronto, Canada. [8]Department of Laboratory Medicine and Pathobiology, University of Toronto, Toronto, Canada. [9]Division of Neurosurgery, The Hospital for Sick Children, Toronto, Canada. [10]Ontario Institute for Cancer Research, Toronto, Canada. [11]The Lunenfeld-Tanenbaum Research Institute at Sinai Health System, Toronto, Canada. [12]Department of Computer Science, University of Toronto, Toronto, Canada. [13]CIFAR Multiscale Human, CIFAR, Toronto, Canada. ✉e-mail: shaminiad@gmail.com; trevor.pugh@utoronto.ca; gary.bader@utoronto.ca

To better understand the multicellular patterns present in GSC cultures, we analyzed 17,000 phase-contrast microscopy images obtained from 15 patient-derived GSC cultures grown over 12–16 days. A set of 29 image analysis features, calculated based on pixel distributions, was computed for each image. Unsupervised analysis reveals that these features naturally organize the GSCs along a spectrum that strongly correlates with a neurodevelopmental-to-injury response gradient and multiple brain cell type and inflammatory gene expression signatures[3,4,7,19–23]. Further, specific image features correlate with transcriptional states across multiple cell confluency levels. This relationship will be useful in developing improved GSC screens where we can identify how key aspects of GSC biology change in response to perturbations by imaging cultured cells on plates.

## Results

### GSCs show diverse morphometric and multicellular patterns in culture

To measure cell organization in 2D cell cultures, we used phase-contrast light microscopy to image GSC cultures from 15 patients, from day 1 until confluency (up to 16 days) at 4 or 12 h intervals. This resulted in a database of 17,601 images (Supplementary Fig. 1A), previously used only for overall cell confluency measurements to develop controls in a chemical screen[15]. However, we observed strong variation in intensities and spatial distributions of pixels across time and confluency levels within these images, suggesting that the images contain additional information related to the spatial organization of the cells (Supplementary Fig. 1B). Manual inspection of image patterns identified rich texture, directionality, cell shape and cell composition information within the images. The most visually prominent pattern variation appears from multicellular structure formation over different confluency levels (Fig. 1 and Supplementary Fig. 1B). For example, we observed in some samples (G564; Fig. 1A, G549, G799, G800; Supplementary Fig. 1B) that cells have a tendency to align with their longitudinal axis against each other, exhibiting anisotropy (a term used to describe directionality in patterning), whereas in other samples we found isotropic (used to describe more uniform orientation or aggregation) patterning of cell clusters (G523; Fig. 1A, G566, G583, G729, G837, G851, G861, G876, G885, G895; Supplementary Fig. 1B). We also observed differences in the spatial geometry of cells (how cells relate to each other in space). For instance, cells from some patient samples (e.g., G837) organize themselves in a 2D layer, exhibiting efficient space utilization with little cell-cell overlap and no visible intercellular gaps (Fig. 1B). This appears to be an inherent property of these cells and not due solely to confluency effects, as the property is visible at multiple confluency levels. However, cells from other samples, such as G876, grow in a largely overlapping manner, making extensive membrane projections that overlap nearby cells (Fig. 1B).

To quantify these visual pattern descriptions, we used CellProfiler to compute diverse pixel composition and pixel-pixel spatial relationship features in an image[18]. CellProfiler implements two classes of whole image feature extraction algorithms: gray-level co-occurrence matrix (GLCM) and granularity spectrum (Supplementary Notes 1 and 2, Materials and methods). Briefly, the GLCM is a symmetric frequency matrix derived from the raw pixel matrix of an image and represents the frequency distribution of

pixel-pairs (Supplementary Note 1). From this frequency distribution, descriptive (i.e., mean, variance, correlation) and spatial (i.e., homogeneity, entropy, and contrast) patterns of pixel-pair relationships across an image are calculated[24]. We also included the granularity spectrum, which captures the size range of structures in an image (Supplementary Note 2). This results in a total of 29 features representing pixel distribution patterns within an image. Plotting feature scores over time of cells growing on a 2D surface quantifies the variation manually observed in the images (Supplementary Fig. 2). For instance, examining a feature score plot for sample G566 (red box in Supplementary Fig. 2), we can clearly observe that the "Granularity 1" feature decreases over time, whereas "Granularity 7" increases over time, suggesting that smaller structures decrease, whereas larger structural elements are being formed (as cells form collective clumps as they grow). Overall, we observe a rich diversity of spatial pixel variation, accompanied by multicellular patterns in our cell culture images.

### Unsupervised factor analysis uncovers spatial patterns in GSC images that correlate with gene expression signatures

We next asked if the multicellular patterns we observe in our images correlate with known biological programs, such as cell type or state. To quantify the pattern variation at multiple confluencies, we divided the data profiled using CellProfiler into different cell densities. We binned these images over nine confluency levels to capture the changes in multicellular organization over varying cell densities as well as to ensure that technical plate effect was minimized (materials and methods, Supplementary Fig. 3A). For each image, we computed a feature vector using the 29 image features described above, normalized by confluency level to control for cell density. We then performed principal component analysis (PCA) over each confluency level. We found that principal component one (PC1) and two (PC2) carried the bulk of the variance (43–63% total, across confluency levels) (Supplementary Fig. 3B). We next asked if the PCs correlate with known biological signals. We analyzed bulk RNA-seq datasets matched with our 15 GSC samples. We scored each RNA-seq sample using 111 gene signatures from human fetal and adult brain cell types and GBM tissues from a range of studies (Supplementary Table 1, Materials and methods[3,4,7,19,22,23,25]). To correlate our image feature vectors captured by image, and our 111 gene signature vectors calculated by transcriptome, we computed the mean PC scores by sample for each principal component and confluency level (materials and methods, Supplementary Fig. 3C). We then computed the Pearson correlation coefficient for each averaged component with each of the 111 gene signatures. Both PC1 and PC2 show biological correlation across confluency levels, though PC2 has a more generalizable association with gene expression (Fig. 2A, B, Supplementary Fig. 4, and Supplementary Table 2). PC2 correlates with signatures representing OPCs, NSCs, neurons and astrocytes, and anti-correlates with signatures representing microglia, mesenchymal and injury response phenotypes (Fig. 2 and Supplementary Table 2). Thus, GSCs in culture exhibit spatial patterning that correlates with gene expression programs, with neurodevelopmental and mesenchymal/injury-response biological programs showing distinct geometric patterns well separated in PC space.

**Fig. 1 | Diverse multicellular spatial patterning of patient-derived glioma stem-like cells in culture.**
**A, B** Representative images of glioma stem-like cells exhibiting rich multicellular spatial patterning. Scale representation = 50 microns for both x and y-axis bars. **A** Cells in the left image show directional packing (anisotropy) as compared to cells in the right image which show cell aggregates packed in more uniformly distributed orientations (isotropy). **B** Cells in the left image show efficient space utilization where cells organize themselves with cell membranes aligned next to each other compared to cells in the right image where packing and space utilization results in overlapping cellular processes.

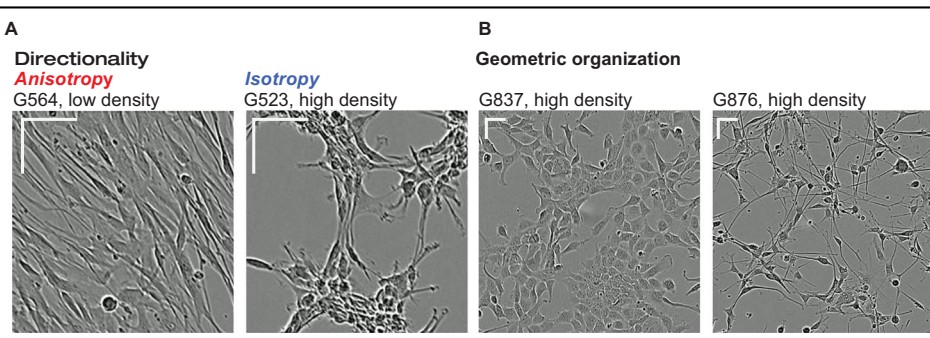

A

**Directionality**
*Anisotropy*
G564, low density

*Isotropy*
G523, high density

B

**Geometric organization**
G837, high density

G876, high density

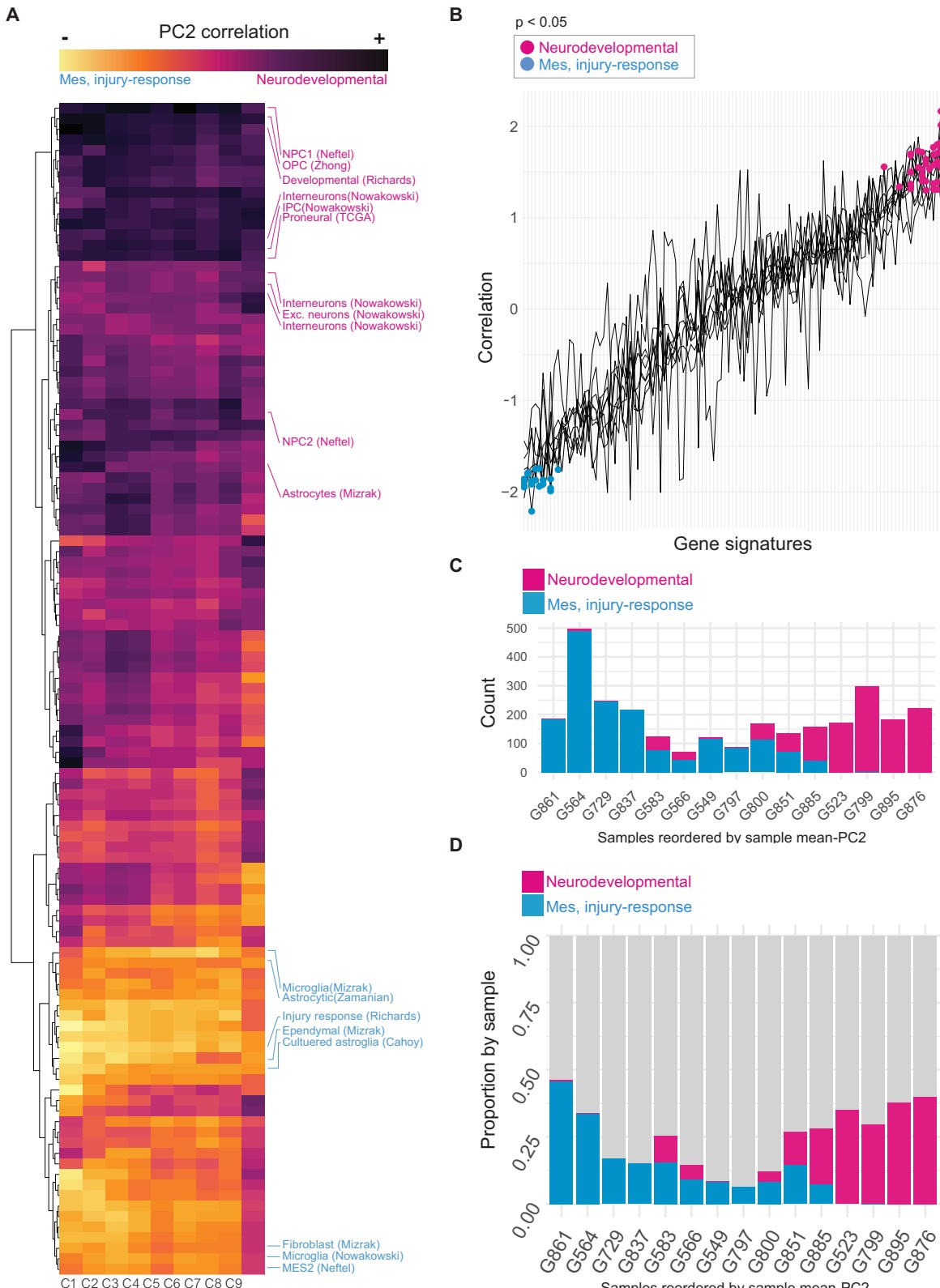

**The GSC multicellular spatial pattern to gene expression correlation is maintained across all growth phases, but relevant image features change over time**

We next interpreted the image features correlated with the neurodevelopmental and mesenchymal/injury-response transcriptional gradient. Overall, these image features show broad patterns that change with cell density (Supplementary Fig. 5). For instance, granularity spectrum values increase as cell density increases (Supplementary Fig. 5A). This trend is expected as at lower cellular densities there are more single or smaller clusters of cells. The GLCM derived image features show a more complex relationship with confluency

**Fig. 2 | Image analysis features correlate with gene signatures. A** Heatmap of correlation scores between image-derived mean-PC2 scores for 15 samples, from each confluency level (columns) and GSVA scores for matched 15 samples for each of the 111 gene signatures. Select gene signatures are labeled and colored by their major biological category (magenta - neurodevelopmental/high PC2, blue - mesenchymal-injury-response/low PC2). The nine confluency levels (columns) are labeled as C1–C9 at the bottom of the heatmap. Heatmap is a z-normalized color map for high and low Pearson correlation scores. **B** The column vectors from heatmap in (**A**) plotted against gene signatures (one line plot per confluency level). The y-axis represents the Pearson correlation between the mean-PC2 vectors for 15 samples and the GSVA scores derived from matched samples, for each of the 111 gene signatures from bulk gene expression datasets (x-axis of the heatmap in panel (**A**). The x-axis represents 111 gene signatures used to compute the GSVA scores and ordered by the correlation values (y-axis of the heatmap in panel **A**). Gene signatures with statistically significant correlations ($p < 0.05$) are colored by the major biological category they represent (magenta = neurodevelopmental and blue=-mesenchymal, injury-response). **C** Images from the top and bottom 25% of PC2 are represented by their sample count (**C**) and fraction (**D**). For panel (**C**) the y-axis represents the number of images whereas in panel (**D**) the y-axis represents the fraction of the number of images present in the top or bottom 25% of PC2 per sample as a fraction of the total number of images per sample. For both (**C, D**), samples are ordered by their mean-PC2 score along the x-axis and colored by the major biological category (magenta = neurodevelopmental and blue = mesenchymal, injury-response). For **A, B**, gene signatures: $n = 111$ and confluency groups: $n = 9$. For **C, D**, sample size: $n = 15$ biologically independent samples.

levels, with some features positively or negatively correlated with cell density and others showing a nonlinear relationship (Supplementary Fig. 5B, C).

These variations in image features were observed along PC2, distributing images along the neurodevelopmental-mesenchymal/injury-response gradient (Fig. 3). Despite maintaining a consistent overall correlation with the neurodevelopmental and mesenchymal/injury-response gradient across confluency levels, the image features driving PC2 variance change over cell growth phases, from low to mid to high confluency (Fig. 3A). For instance, small granularity features are consistently associated with lower cell densities (Fig. 3A). However, within these lower cell densities, the PC2 loadings are higher for the low granularity features with respect to the neurodevelopmental GSC culture images, indicating smaller structures within these images, in contrast to images from the mesenchymal/injury response GSC culture images of similar densities (Fig. 3A, Supplementary Fig. 6A, and Supplementary Table 3). In contrast, PC2 loadings for the mid granularity features are stronger in the low density mesenchymal/injury response GSC culture images (Fig. 3A, Supplementary Fig. 6B, and Supplementary Table 3). GLCM-derived features also show a clear biological association with the neurodevelopmental-mesenchymal/injury-response gradient. For instance, the informational measure features (IMC1 and IMC2), both measuring the mutual information along the horizontal and vertical image axes (Supplementary Note 1), are the strongest PC2 feature loadings but exhibit varying influences at different confluencies and NIR scores (Fig. 3A). IMC1 is strongly correlated with the mesenchymal/injury response signatures at the low-mid confluency levels whereas IMC2 exhibits a stronger correlation with the neurodevelopmental samples at high confluency levels (Fig. 3B and Supplementary Table 3). This reflects the symmetry of multicellular structures at two different density ranges and shows how neurodevelopmental cell states grow in more symmetric multicellular patterns than mesenchymal/injury-response cell states. Another example is the inverse difference moment, which measures homogeneity by penalizing high contrast areas. It is associated with neurodevelopmental images at mid-high confluency levels (Fig. 3A, C and Supplementary Table 3). The correlation feature, measuring similarities in pixel neighborhood patterns in the image, is also more strongly associated with neurodevelopmental images at mid confluencies (Fig. 3D and Supplementary Table 3). Differences in pixel-pair values along different axes and directions can be attributed to variation in the sharp boundaries or the nature of pixel-intensity shifts that form as cells meet each other along their membranes, overlap cellular processes or exhibit anisotropy, forming variable local light-dark contrast patterns. The data indicates a more consistent cell population patterning within the neurodevelopmental images compared to the mesenchymal and injury-response samples and this may relate to consistent overall cellular composition, resulting in cells maintaining similar textures even as they grow into larger cellular networks. Overall, image analysis features highlight organizational differences along the neuro-developmental-to-mesenchymal/injury-response biological gradient that are preserved despite the changing spatial organization patterns of cells in culture during growth.

## Predicting neurodevelopmental-injury response score from image features

We next trained a predictive model to map GSCs along the neurodevelopmental-injury response (NIR) transcriptional gradient using only 2D culture image features. Pooling all confluency levels, after group-wise feature normalization, we trained all image data points to predict the first principal component after applying PCA to the 111 GSVA-derived gene signature scores (Fig. 4A)[26]. We tried Ridge, Lasso and Elastic Net regularized regression (Fig. 4B). The Lasso regularization model gave the best overall performance (Fig. 4B) ($R^2 = 0.788$, $p$ value = 1e-05). We next tested using the trained lasso model by predicting the NIR gradient for four new samples that had matched phase-contrast image and bulk gene expression data. While not significant, likely due to variation among a low number of samples, model prediction was correlated with ground truth values and agreed on the overall trend of high or low on the NIR gradient ($R^2$ value of 0.288, $p$ value = 0.464) (Fig. 4C). The features with the most weight contribution in the Lasso model are Angular second moment, Granularity 5, Granularity 6, and Contrast. These are similar to our PC2 loadings, averaged across confluency levels. For example, the Angular second moment highlighted by our Lasso model with a positive weight is amongst the top five of our averaged PC2 positive loadings (Supplementary Table 3) and the granularity features dominating our Lasso regression model with negative weights are amongst the PC2 negative loadings (Supplementary Table 3). The Lasso model outperformed a targeted approach when training a linear model on two top image-features (topmost and bottommost highest PC2 loaded features: Informational measure 1 and Granularity 13; $R^2 = 0.368$, $p$ value = 0.0165) (Fig. 4D). Overall, phase-contrast image features can be used as a predictive marker for GSC transcriptional state.

## Discussion

Microscopic images of cells growing over time on 2D plates are a rich source of biological information, but are traditionally used to extract single values, such as cellular growth rate, for analysis. The multicellular spatial patterns visible in these images are defined by cell-type orientation, space utilization, cell contacts, and cell shape. Here, we show that these patterns vary along temporal and biological axes. In particular, cultured glioblastoma stem cells form patterns that strongly correlate with a transcriptomic gradient expressing neurodevelopmental pathways on one side and mesenchymal and injury-response/inflammation pathways on the other[4]. Mesenchymal/injury response GSCs show increasingly more diverse multicellular patterns, while neurodevelopmental samples exhibit more stable patterning as they grow. We also find that spatial patterning varies with time and yet maintains a strong correlation with the GSC transcriptomics gradient, suggesting that as spatial structures evolve, they are constrained by underlying biological programs.

A key open question is how single-cell morphometric features relate to the higher-order multicellular patterns we studied. Is there more spatial pattern information available at the multicellular level than can be discerned from studying individual cells? Unfortunately, we couldn't address this as we had relatively few single-cell examples to study in our data. Further, extracting high-quality single-cell shapes from multicellular neighborhoods

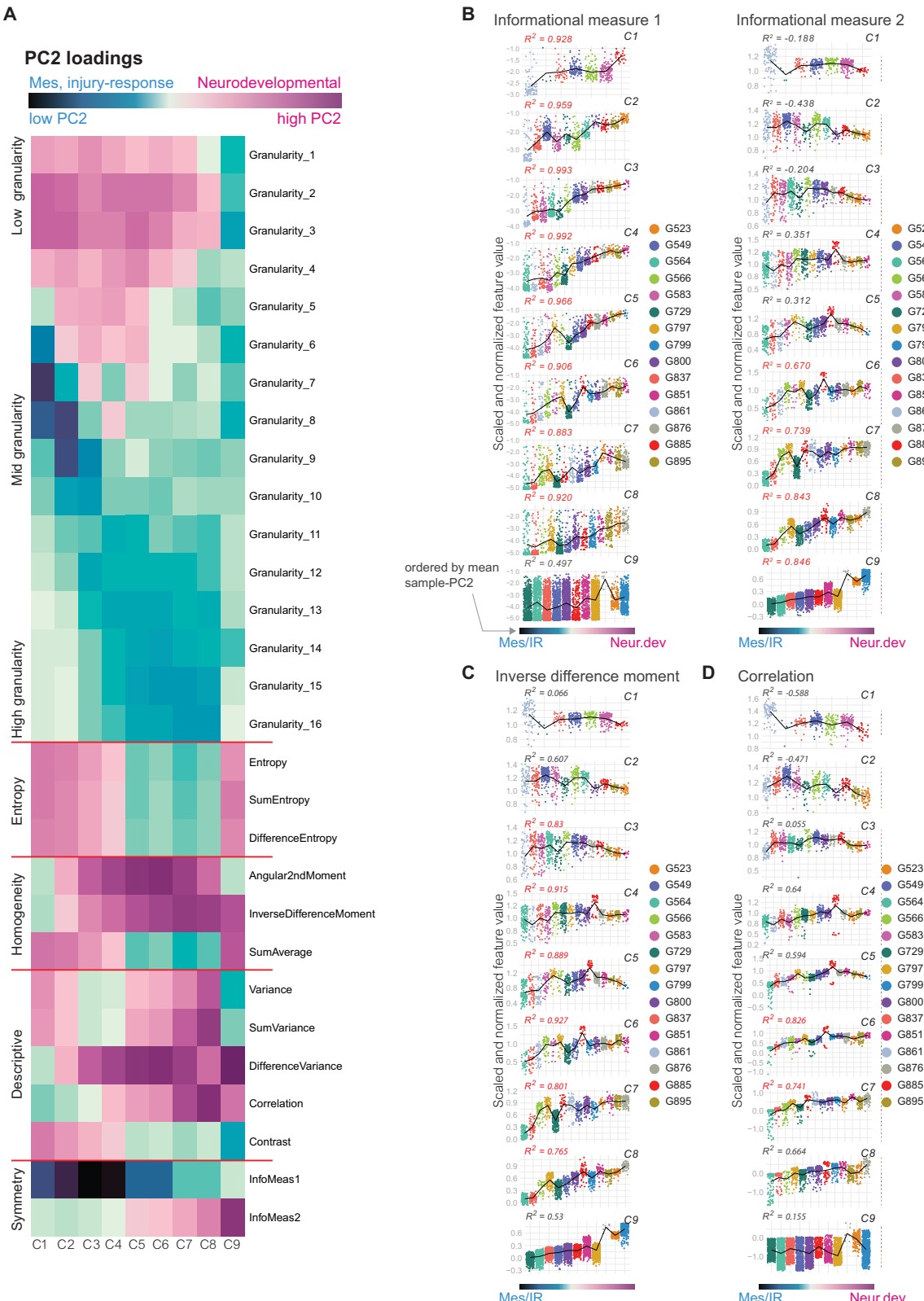

**Fig. 3 | Identifying image analysis features that best correlate with GSC neuro-developmental/injury response transcriptomic gradient. A** A heatmap of the PC2 loadings for all 29 image features from PC2. Blue indicates low values correlated with mesenchymal, injury response GSCs and magenta indicates high values, correlated with neurodevelopmental GSCs. Related features are grouped (group labels on the left). **B**–**D** Representative distribution plots of images grouped by sample with a line plot showing the trend. For **B**–**D**, the y-axis represents the scaled and normalized values for each feature represented in the panel (see Materials and methods) and the x-axis represents samples ordered along the x-axis by their mean-PC2 value. Colors represent images from each sample. For panels **B**–**D**, the $R^2$ value is colored red if $p < 0.01$. For **A**, confluency groups: $n = 9$, Image features: $n = 29$. For **B**–**D**, biologically independent sample size within each confluency group: C1:$n = 7$, C2: $n = 11$, C3: $n = 13$, C4: $n = 14$, C5: $n = 15$, C6: $n = 15$, C7: $n = 14$, C8: $n = 13$, C9: $n = 11$.

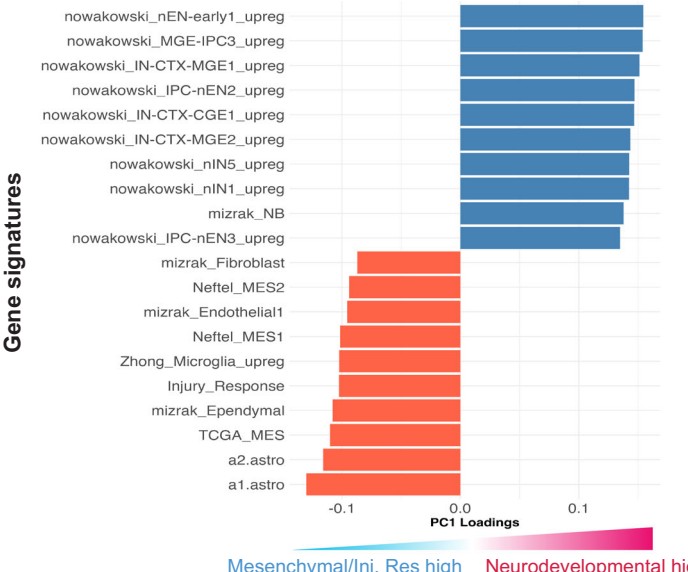

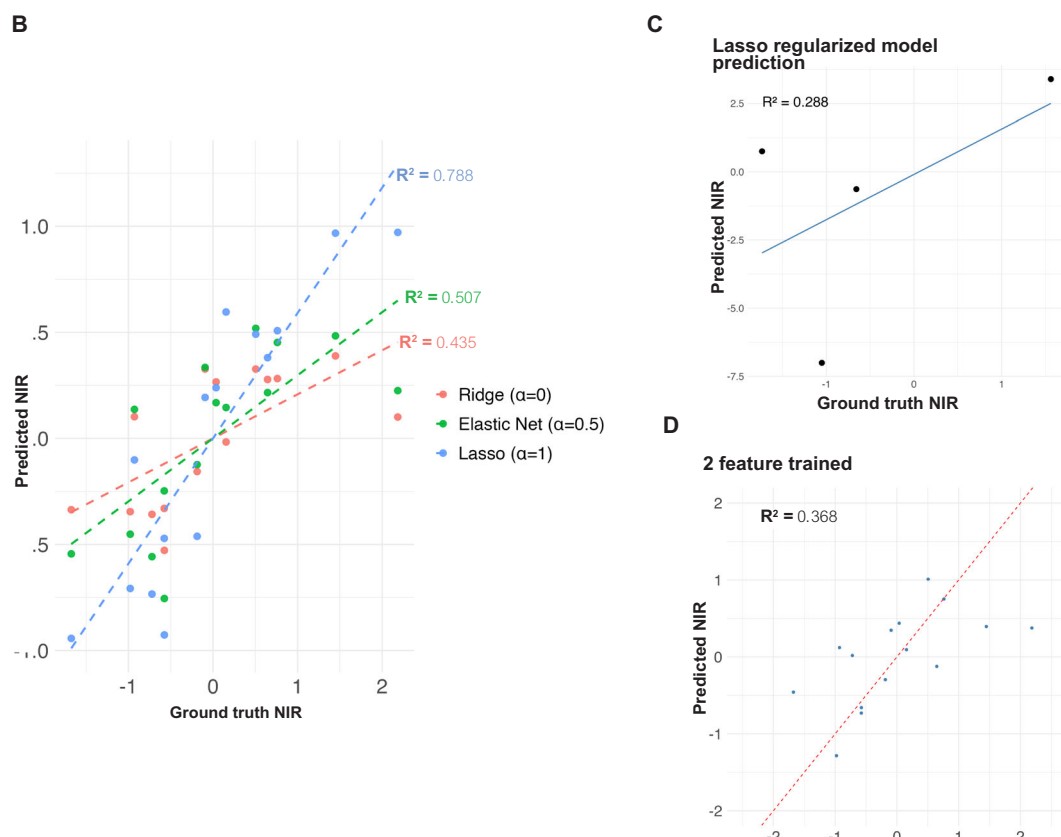

**Fig. 4 | Mapping the neurodevelopmental-injury response gradient. A** The first principal component (PC1) of the GSVA score matrix shows variance corresponding to the neurodevelopmental/injury response transcriptional gradient and is thus designated the "NIR gradient". **B** Performance of regression models using 29 image features as predictors and gene expression derived neurodevelopmental-injury response gradient as the response variable. Each colored line indicates a different regression model and each line is composed of 15 sample points used to train the model. **C** Performance of a Lasso regularized linear model in predicting the NIR gradient using image features for four independent samples not used to train the Lasso model. **D** Performance of a two-feature trained linear model in predicting the NIR gradient on 15 samples. For **B–D**, the y-axis represents the neurodevelopmental-injury response gradient as predicted by the model(s), while the x-axis shows the computed gradient, derived from the first principal component (PC1) of the GSVA scores based on 111 gene signatures. Source data for Fig. 4A and Supplementary Table 5. Source data for Fig. 4B–D and Supplementary Table 6. For **B**, sample size: $n = 15$, for **C**, sample size: $n = 4$ biologically independent samples, and for **D** Sample size: $n = 15$ biologically independent samples.

is challenging for various reasons, such as cellular overlap and difficulty resolving extended cellular membrane protrusions from each other, even when using expert manual annotation. Improved cell segmentation and imaging methods, including those that consider more detailed time series to resolve overlapping cells, cellular dyes and perhaps spatial transcriptomics, will be required to extract enough single-cell images from multicellular structures in 2D culture images to learn these multiscale relationships[27]. Single-cell transcriptomics may provide more information, and can be pseudobulked to compare with our work with bulk RNA-seq.

Extending our 2D study to other models, such as live and fixed tissue slices imaged with diverse microscopy technologies, as well as three-dimensional systems, such as organoids, is expected to provide richer information than can be extracted from the brightfield phase-contrast images we analyze here. However, our general analysis approach using whole image-based features considering all structures in view should be generally compatible with other imaging systems and models.

Our Lasso model can predict the transcriptional phenotype of GSCs from phase-contrast image features. Predicting the phenotype of patient-derived cell lines without the need for sequencing assays can facilitate high-throughput screening and help identify, for example, chemical compounds that perturb cells and their position on the neurodevelopmental-to-injury-response gradient.

Ultimately, we hope that further study of relationships between images and functional genomics data will identify gene expression programs that are correlated with specific multicellular structures, extending our interpretability capability to identify causal biological relationships directly from cell images. This will be useful to improve our understanding of disease models and develop new therapies that consider the architecture of growing multicellular systems.

## Materials and methods

### Glioblastoma stem cell culture

Fresh tumor samples were obtained from patients during operative procedures following informed consent. All experimental procedures were performed in accordance with the Research Ethics Board at The Hospital for Sick Children (REB1000025582 and REB0020010404), the University Health Network, the University of Calgary Ethics Review Board and the Health Research Ethics Board of Alberta, Cancer Committee and Arnie Charbonneau Cancer Institute Research Ethics Board (REB HREBA-CC-160762)[4,15]. All ethical regulations relevant to human research participants were followed. Authentication of cell lines was done with PCR using a panel of polymorphic markers. The lines were matching the patients they were derived from. All lines were tested for mycoplasma contamination and were found to be free of contamination. Adherent glioma stem cell (GSC) lines were propagated in adherent format on tissue culture plates coated with poly-L-ornithine and laminin. Cultures were maintained in serum-free neural stem cell self-renewal medium composed of NeuroCult NS-A Basal Medium supplemented with 2 mM L-glutamine, N2 and B27 supplements, 75 µg/mL bovine serum albumin, 10 ng/mL recombinant human epidermal growth factor (EGF), 10 ng/mL basic fibroblast growth factor (FGF), and 2 µg/mL heparin. All experiments were performed using cells between passages 8 and 12.

### Phase-contrast imaging

The phase-contrast image data were collected using an established protocol[15]. Briefly, two thousand GSCs, maintained between passages 8 and 12, were plated adherently in 384-well CELLBIND plates (Corning) and imaged using the Incucyte Zoom™ live cell imaging system (Essen Biosciences). Cells were imaged with a Nikon 10x objective using phase-contrast mode every 4–8 h until the experimental endpoint of a confluent plate (12–16 days). Culture media was changed every 5 days. Images used in this study were from wells grown untreated (stem cell media only) and are $1266 \times 944$ with a pixel size of 1.34 um/pixel. A total of $n = 17{,}601$ images were captured for the discovery analysis from $n = 15$ patient-derived GSCs.

For the validation cohort using regression analysis, $n = 4$ patient-derived GSCs were added, which resulted in a total of $n = 6670$ images.

### Frame extraction before image analysis

Images were saved from the microscope as time-lapse videos (.mp4 video files), and openCV was used to extract image frames from those videos. Each phase-contrast image was saved as a Tag Image File Format (TIFF) file.

### Whole image analysis—Ilastik and CellProfiler analysis pipeline

**Mask generation with Ilastik.** For each TIFF image, we used the ilastik(v1.4.0) software (pixel classification module) to generate background segmentation masks that exclude any "acellular" regions of the plate/image during downstream computation[28]. Every second image until $t = 24$ h and every sixth image thereafter until $t = 168$ h were used for training. Each sample dataset was processed independently with its own training run, to account for sample-level batch effects. Two classifiers, background and foreground, were created within each Ilastik session to separate plate pixels from cell pixels. A single colored stroke, with stroke width ranging from 1 to 3, was used to mark the background/plate area and a single stroke with a different color was used to mark the cell-occupied regions. Any debris present was included as part of the foreground. This process was repeated for a sampling of images (every 3rd–5th image). If necessary, training was continued using additional unused images. Each image used in training only received 1–2 training strokes per classifier group per image. After training, the ilastik pixel classifier for each sample dataset was run on a computer cluster using all images. The background and cell masks generated were used downstream for image data extraction using CellProfiler's built-in image analysis algorithms[18].

**Image preprocessing and image analysis.** First, raw images were screened using the CellProfiler(v4.0.7) "MeasureImageQuality" module to remove poor quality images[24]. After trying out several of the module's parameters, we found that the "PowerLogLogSlope" (PLLS) function was sufficient to remove aberrant images[29]. The images removed by PLLS were manually verified to ensure that they were low quality (e.g., blurry or corrupted by noticeable artifacts, such as pipette plastic objects, well bottom scratches) and no exceptions were found. After QC, 435 images from G523, 1401 images from G549, 1449 images from G564, 456 images from G566, 456 images from G583, 1433 images from G729, 1330 images from G797, 892 images from G799, 1323 images from G800, 1417 images from G837, 477 images from G851, 446 images from G861, 472 images from G876, 530 images from G885, and 441 images from G895 were retained for downstream analysis. For the validation set, post-QC processing resulted in 456 images from G411, 333 images from G620, 493 images from G637, and 74 images from G683. The foreground masks, representing cellular regions, were used to calculate the area of cells occupying each image. Following this, the "MeasureTexture" and "MeasureGranularity" modules were used to extract gray-level co-occurrence matrix pixel-pair distributions and the granular or size distribution patterns from these images using 29 image analysis features, with default settings (Supplementary Notes 1 and 2). The "Measure-Granularity" algorithm uses 16 different pixel sizes of openings/closings or structuring elements as its default. These were used to analyze the granularity of each image and its relative size. The "MeasureTexture" module measures the GLCM, which consists of 13 features, measuring various aspects of pixel-pair spatial distribution patterns, using the gray scale co-occurrence matrix. For the GLCM features, a scale factor from 0 to 3 was used. The scale factor refers to the directionality of the GLCM matrices ("north", "south", "east", "west") and is explained in Supplementary Note 1). These four directions perfectly correlate with each other; thus, we included only a single scale (direction = "east") from our data analysis (scale = 0 in CellProfiler).

**Whole image analysis—data analysis using R**. All data processing and statistical analyses were conducted in R (4.4.2) within an active renv environment to support reproducibility. Major Bioconductor and CRAN packages used BiocManager (1.30.25), SummarizedExperiment (1.36.0); GSVA (2.0.6), and GSEABase (1.68.0) for gene-set variation analysis; PCAtools (2.18.0) for principal-component computations and visualization; glmnet (4.1-8) for ridge and elastic-net regression; dplyr (1.1.4) and tidyr (1.3.1) for data wrangling; reshape2 (1.4.4) for reshaping data matrices; ggplot2 (3.5.1), ggpubr (0.6.0), and ggbiplot (0.6.2) for data visualization and figure assembly; and pheatmap (1.0.12) for heatmap generation. Each package version was recorded in the project's renv.lock file to support consistency across analyses.

**Normalization**. Image features were normalized by sample. The data matrix is composed of rows representing individual images and columns representing one of the 29 image features. Each feature vector or column was multiplied by a diagonal matrix, where the diagonal represented the scaling factor to transform each feature vector as a fraction of the maximum feature value.

$$Ic = \text{diag}(a_1, a_2, a_3 \ldots., a_f)[f_1, f_2, f_3, \ldots f_n]^T$$

I = Image matrix, c = image column, $a_f$ = maximum value of pixel feature in each column c, $f_n$ = feature vector

Following this, z-normalization was applied by row to obtain a normalized feature vector for each image. Batch correction and data binning into nine confluency levels Principal component analysis (PCA) over all images is strongly driven by the time-point at which the images were taken and the sample (batch effect). To mitigate batch effects, we grouped the images by time-point, to ensure that images from the same plate and time-point were grouped together. Next, we formed batches of images based on the mean total cell area per image while grouped by time-point for each sample set to form different confluency levels. Altogether, we separated the images into nine confluency levels (Supplementary Fig. 3A). The nine confluency levels from C1 to C9 were composed of n images as follows: 487, 747, 803, 868, 822, 802, 1104, 1302, and 5974. Some overlap between levels is caused by technical or biological effects, such as variable sample seeding density or larger cell size, but overall, the binning procedure was successful at removing major technical effects.

**Principal component analysis of whole image analysis**. PCAtools version 2.18.0 with default parameters was used for PCA dimensional reduction of all images (operating on image features).

**Bulk RNA-seq analysis**

Bulk RNA sequencing data were previously generated from glioma stem-like cell (GSC) cultures and analyzed using gene set variation analysis (GSVA)[26]. GSC-relevant gene signatures were curated from Richards et al. (Supplementary Table 4)[4]. Cells were maintained between passages 8 and 12 and cultured to ~70–80% confluency prior to collection for bulk RNA sequencing. The details of library preparation, read alignment and other preprocessing steps previously generated are outlined below as described in the original publication[4].

**Library preparation and sequencing**. Total RNA was isolated from frozen GSC pellets using the AllPrep DNA/RNA Universal Kit (QIAGEN). Strand-specific RNA sequencing libraries were prepared from 500 ng of total RNA following poly(A) transcript enrichment using the NEBNext Poly(A) mRNA magnetic isolation module (E7490L, New England Biolabs). Libraries were quantified using the Qubit dsDNA High Sensitivity Assay (Thermo Fisher Scientific). Cluster generation and sequencing were performed on the Illumina HiSeq 2500 platform using indexed lanes and v4 chemistry, in accordance with the manufacturer's protocols.

**Read alignment and preprocessing**. Strand-specific, paired-end 75 bp reads were aligned to the human reference genome (hg38) using STAR (v2.4.2a), with gene annotation based on the UCSC reference from the Illumina iGenome resource. Raw gene-level counts were obtained from the STAR ReadsPerGene output. Genes with fewer than five total counts across all samples were excluded from downstream analyses.

**Data transformation and pathway-level analysis**. Normalization and variance stabilization were performed using DESeq2 (v1.22.2), which was also used to estimate sample-specific size factors. Batch correction was applied to account for both technical and biologically relevant covariates in the model[30]. GSVA (v1.30.0) was used to quantify pathway and gene signature activity across samples[26]. Gene signatures were compiled from multiple published sources capturing neural progenitor, neuronal, oligodendrocyte progenitor, oligodendrocyte, astrocytic, mesenchymal, immune, tumor-associated, and injury-response/developmental programs[3,4,7,19,20,22,23,25]. Gene signatures showing downregulated patterns were excluded to reduce redundancy with corresponding upregulated signatures. After filtering, a total of 111 gene signatures were retained for downstream analyses.

**Pearson correlation analysis between the whole image and the bulk RNA-seq datasets**

For each confluency level, we computed the Pearson correlation coefficients between the normalized image features and the 111 gene signatures obtained from bulk RNA-seq. First, the mean PC scores were calculated for each GSC for all 15 samples, resulting in a vector of 15 mean PC scores, which were then correlated against each of the 111 gene signature vectors, where each vector comprises the GSVA scores for a single unique signature across 15 GSCs. PC signs were aligned to a common direction to support meaningful biological comparisons across confluency levels and gene signatures.

To obtain the top and bottom 25% images along PC1 and PC2, first, the PC1 and PC2 scores for all images were z-normalized by confluency levels. Next, z-normalized matrices were concatenated to form a single dataframe and ranked by PC2 scores. The top 25% and bottom 25% ranked images were subsetted to calculate the proportion of sample contribution. Where samples were indexed along the x-axis, order was determined by mean-PC2 score. $P$ values obtained from the corr.test output were considered significant when values fell below 0.01. The correlation coefficients and $p$ values can be found in Supplementary Table 5.

**Linear regression modeling**

**Reduced gene signature dimensionality using principal component analysis**. To reduce dimensionality and identify dominant expression patterns, principal component analysis (PCA) was applied to the GSVA score matrix (described above) using base R's prcomp() function. The first principal component (PC1), explaining the largest proportion of variance, captured a neurodevelopmental-to-injury-response (NIR) gradient, with high PC1 scores aligning with neurodevelopmental phenotypes and low scores with mesenchymal and injury-response states (Fig. 4A). PC1 was thus designated the "NIR gradient" for all subsequent analyses.

**Image feature aggregation**. Phase-contrast image-derived features were computed for each glioma stem cell (GSC) sample across multiple confluency conditions. To obtain a single representative feature vector per sample, features were averaged across all confluency levels (after normalizing within confluency levels). This aggregation was applied to both the original dataset and four additional GSC datasets containing matched imaging and gene expression profiles.

**Model training and prediction**. Linear regression models were implemented using base R's lm() function, with the NIR gradient (PC1 scores) as the response variable and image features as predictors. To account for

potential multicollinearity and enable feature selection, three regularized linear models—Ridge (alpha = 0), Lasso (alpha = 1), and Elastic Net (alpha = 0.5)—were trained using the glmnet package in R. All models were trained on the original dataset of averaged image features (across samples) and used to predict NIR scores in the four new GSC datasets.

**Feature-reduced modeling.** To assess the feasibility of using a small feature set to predict NIR gradient position, a linear regression model was constructed using only the two image features with the highest positive and negative loadings on the second principal component (PC2) of the feature matrix: Granularity 13 and Informational Measure 1 (Supplementary Table 3).

**Implementation.** All data preprocessing, modeling, and visualization steps were performed in R using base packages and glmnet. Model performance was evaluated using $R^2$ values and predicted-versus-actual plots.

### Statistics and reproducibility
**Study design and replicates.** The primary image dataset was generated from 15 biologically independent patient-derived GSC lines (biological replicates). For each line, longitudinal phase-contrast images were acquired temporally from seeding to confluence, yielding 17,601 images that constitute technical replicates (multiple time points over a single field per line per well). For external evaluation of the imaging-to-transcriptome mapping, four additional independent GSC samples with matched imaging and gene expression were used. The images from these four samples were acquired similarly to what was described before. Bulk RNA-seq was performed once per biological replicate ($n = 15$), with four additional biological samples ($n = 4$) for external evaluation.

**Reproducibility.** All pipelines (CellProfiler/ilastik settings and R analysis scripts) are fully versioned and provided, enabling end-to-end reproduction from raw images/masks to figures and statistics.

**Statistics.** All statistical analyses were performed in R. Relationships between image-derived features and gene expression data were evaluated using simple linear regression and Pearson correlation. Model performance and associations were reported as $R^2$ and $p$ values, with $p < 0.01$ considered statistically significant.

### Inclusion and ethics statement
The study was designed to address biological and technical questions using in vitro models. No human participants or animals were directly involved, and no demographic, race, ethnicity, or other socially defined variables were collected or analyzed. All analyses were conducted on de-identified data in accordance with institutional and ethical guidelines.

### Reporting summary
Further information on research design is available in the Nature Portfolio Reporting Summary linked to this article.

### Data availability
The Cellprofiler image feature output for each image, GSVA scores for 111 gene signatures from 15 matched bulk gene expression samples and all images, Supplementary Tables 1–6 and related data can be found on Zenodo[31]. Source image for Fig. 1 and Supplementary Fig. 1 can be found in the zipped folder deposited onto the Zenodo data repository, under the subdirectory "datasets_final/datasets_images_ilastik_training_and_manuscript_figures/fig_original_images"[31]. Source data for Fig. 2C, D, Supplementary Figs. 2, 3, and Supplementary Figs. 5, 6 can be obtained from the following source subdirectory deposited onto the Zenodo data repository - "datasets_final/manuscript_analysis_data/cellprofiler_output"[31]. Source data for Fig. 2A, B and Supplementary Fig. 4 can be accessed from Supplementary Table 2. Source data for Fig. 3 can be accessed from Supplementary Table 3. Source data for Fig. 4A can be accessed from Supplementary Table 5, and Source data for Fig. 4B, C can be accessed from Supplementary Table 6[31]. Previously published bulk RNA-seq data that were re-analyzed in this study are available from the following sources: EGAS00001003070 and EGAS00001004395 through the European Genome-Phenome Archive repository in the form of FASTQ or BAM files.

### Code availability
All code, CellProfiler pipelines, and ilastik pipelines used in the manuscript are deposited on GitHub at the following link https://github.com/BaderLab/phase_contrast_collective_organization_GSCs_incucyte_August2025. A copy of all code used for analysis has been uploaded onto Zenodo and represents the analysis version for the submission/preparation of this manuscript[31]. The gene set processing pipeline is publicly available at: https://github.com/BaderLab/owen_su2cproj/blob/integrated_analyses/RNA/preprocessing/preprocess_GeneSets.Rmd.

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

## Author contributions

S.A. conceived the study. S.A., T.J.P., and G.D.B. designed the study with methodological development by S.A. and G.D.B. P.S., M.M.K., N.I.P., and F.J.C. designed, developed, and executed the experimental work for cell culture and imaging under the supervision of P.P., C.H.A., and P.B.D. All computational analysis was done by S.A., except for the bulk RNA sequencing pipeline from O.W., under the supervision of G.D.B. The manuscript was written by S.A., T.J.P., and G.D.B. The manuscript underwent final revisions with input from all authors.

## Competing interests

The authors declare no competing interests.
