## [Transparent Peer Review file · Communications Biology]

Glioblastoma stem cells show transcriptionally correlated spatial organization

Corresponding Author: Dr Shamini Ayyadhury

Version 0:

Reviewer comments:

Reviewer #1

(Remarks to the Author)

The authors utilize the CellProfiler tool for image segmentation and feature analysis, focusing primarily on phase contrast images of glioma stem-like cells. Their approach characterizes phenotypic patterns through measures of directionality, geometric organization, and pixel density. A key finding is the correlation between these phenotypic traits and mRNA sequencing-derived gene expression signatures, mapped along a neurodevelopmental-mesenchymal/injury response axis. The study's open-source methodology (available on GitHub) offers an accessible framework, which can help to enhance the characterization of patient-derived glioma cells—now widely recognized as workhorse models in glioma research. The work adds depth to the understanding of glioma stem cell phenotypes.

Major points

- 1) The evidence correlation mainly relies on multi-parameter heatmap visualizations. To improve usability, the authors should consider defining an index-based classification system for rapidly mapping glioma stem-like cell lines onto the neurodevelopmental-mesenchymal axis. This would make their approach more accessible and widely applicable.
- 2) With the increasing use of single-cell RNA sequencing (scRNAseq), the authors should discuss how their approach could adapt without further experiments. Addressing the integration of scRNAseq would provide insight into the method's future-proofing and relevance.
- 3) The manuscript discusses the limitations of single-cell morphology analysis, yet Figure 1A appears to emphasize such features. The authors should explain this discrepancy and clarify the role of single-cell morphology in their analysis.

Minor points

Extend the figure legends, particularly for Figure 1, which currently lacks clarification on arrows and markers. Ensure all necessary information is included so the legends can stand alone.

Provide a citation for the CellProfiler tool in the narrative (e.g., on page 4).

Quantify the data in Supplemental Figure 1B, as it is used to introduce 'pixel variation across time' (page 3) but lacks numerical details to support this claim.

Describe the origin of the neurodevelopmental-mesenchymal/injury axis in more detail and cite previous work that supports this concept.

I suggest proofreading the manuscript from a non-expert perspective, minimizing technical jargon or providing clear explanations for specialised terms to enhance readability.

List the exact software or open-source tools used for sequencing and correlation analysis to ensure reproducibility of the methods.

Reviewer #2

(Remarks to the Author)

Summary: In this manuscript, the authors measured features of texture and granularity on phase contrast images of in house patient derived GSC stem cell lines across time points spanning the range of phenotypes seen as cells grow to confluence. A principle component analysis revealed two principle components that explained a large proportion of the variance seen. PC2 was found to correlate with transcriptional state in the lines matching a neuronal developmental phenotype.

Overall comments: The authors have provided novel exploration of a label free spatial analysis framework and evidence that these features may correlate with transcriptional state within GSC lines. Evidence for this result would be significantly strengthened by testing the findings on a validation set, i.e. derive or purchase a new set of lines, collect phase images and predict developmental state via the minimal required measurement set (perhaps informational measure 1 or 2).

Specific comments:

1. Figure 2C and 3B – How was the order of the GSC lines on the x-axis here determined?
2. Figure S2A – Why does the covered area vary so much for many of these lines to the degree that they are often largely overlapping across the confluence levels? The derivation of the confluence levels needs further explanation. Why not have defined cut points of covered area leading to distinct bins of confluence?
3. Figure S2A - Axis labels are too small to read. GSC line – plot color should stay consistent across the plots.
4. Figure 3B - Why does the number of lines differ across the confluence points?
5. Figure 3B - It appears that the GSC line to color pairing differs across the confluence points, this should be corrected. Ideally, we should be able to follow the same GSC line down the column in the same position and color to track how the measure is changing across confluence.
6. Figure 3B – it appears from the dot density comparing down columns, that for many of the GSC lines they achieved the maximum degree of confluence and then remained there for some time. It may be informative to plot these points over time to see if these metrics change over time after confluence is reached.

Version 1:

Reviewer comments:

Reviewer #1

(Remarks to the Author)

I feel that the authors have adequately addressed the reviewers' comments, strengthening the paper by adding validation data, introducing new discussion points, incorporating relevant literature, and providing clearer figure legends.

However, careful proofreading and verification of figure citations are advised; for example, the rebuttal letter refers to Supplementary Figure 7, whereas the supplementary file includes only Supplementary Figures 1–6?

REVIEWER COMMENTS AND RESPONSE

Reviewer #1 (Remarks to the Author):

The authors utilize the CellProfiler tool for image segmentation and feature analysis, focusing primarily on phase contrast images of glioma stem-like cells. Their approach characterizes phenotypic patterns through measures of directionality, geometric organization, and pixel density. A key finding is the correlation between these phenotypic traits and mRNA sequencing-derived gene expression signatures, mapped along a neurodevelopmental-mesenchymal/injury response axis. The study's open-source methodology (available on GitHub) offers an accessible framework, which can help to enhance the characterization of patient-derived glioma cells—now widely recognized as workhorse models in glioma research. The work adds depth to the understanding of glioma stem cell phenotypes.

Major points

1) The evidence correlation mainly relies on multi-parameter heatmap visualizations. To improve usability, the authors should consider defining an index-based classification system for rapidly mapping glioma stem-like cell lines onto the neurodevelopmental-mesenchymal axis. This would make their approach more accessible and widely applicable.

Thanks for this useful suggestion. We have developed a regression model to map glioma stem-like cells (GSCs) onto the neurodevelopmental-injury response (NIR) gradient based on their phase contrast images. We have now included this result in the manuscript (Figure 4A-B).

To build the model, we first applied principal component analysis (PCA) to the gene set enrichment scores computed by gene set variation analysis (GSVA) on our bulk RNA-seq dataset (111 gene signatures). The first principal component (PC1) captures the NIR gradient—high PC1 scores aligned with neurodevelopmental phenotypes, while low PC1 scores correspond to mesenchymal and injury-response states (Figure 4A). Henceforth, we refer to PC1 as the “NIR gradient.” We then pooled all images into one training set by averaging each image feature per sample across all confluency groups. We fit three models (Ridge, Lasso, and Elastic Net regularized regression) using all features on this combined dataset (Figure 4B). We used regularized regression to avoid over-fitting, given that we have more features than samples. The Lasso model achieved the highest predictive accuracy ($R^2 = 0.788$). These results demonstrate that image-derived morphological features can reliably predict gene-expression-defined positions along the NIR gradient.

2) With the increasing use of single-cell RNA sequencing (scRNAseq), the authors should discuss how their approach could adapt without further experiments.

Addressing the integration of scRNAseq would provide insight into the method's future-proofing and relevance.

This is a good point. Our scoring system should be applicable to pseudobulked scRNA-seq data. We have now included this in the discussion section of the manuscript.

3) The manuscript discusses the limitations of single-cell morphology analysis, yet Figure 1A appears to emphasize such features. The authors should explain this discrepancy and clarify the role of single-cell morphology in their analysis.

Apologies for the confusion caused by our Figure 1A design. Figure 1A is intended to illustrate the wide range of cell morphology—from individual cells to multicellular communities—and to explain why we chose to focus on collective organization in this study. Although single-cell morphology is the prevailing metric in most high-throughput screens, new spatial patterns emerge when cells assemble into higher-order structures, which we focus on in our analysis. We actually tried analyzing the data at single cell level, but it was too noisy and we could not include it in the manuscript (data not shown). To maintain focus on multicellular structure analysis, we have removed the single-cell panels from Figure 1.

Minor points

4) Extend the figure legends, particularly for Figure 1, which currently lacks clarification on arrows and markers. Ensure all necessary information is included so the legends can stand alone.

We have extended the figure legends for figure 1 and also reviewed figures 2 and 3 to ensure that legends can stand alone.

5) Provide a citation for the CellProfiler tool in the narrative (e.g., on page 4).

We apologize for this oversight. The citation for CellProfiler has been added.

6) Quantify the data in Supplemental Figure 1B, as it is used to introduce 'pixel variation across time' (page 3) but lacks numerical details to support this claim.

We added Supplementary Figure 2 to quantify how granularity and textural patterns vary over time for all images. Clear patterns are visible that match our visual observations. For instance, in the boxed sample plot in Supplementary Figure 2, G566, we can clearly observe that Granularity 1 decreases with time whereas Granularity 7 increases with time, showing that smaller structures decrease, whereas larger structural elements are being formed (as cells form collective clumps).

7) Describe the origin of the neurodevelopmental-mesenchymal/injury axis in more detail and cite previous work that supports this concept.

We have now included additional content describing this axis and cited the major studies that support this in the introduction.

8) I suggest proofreading the manuscript from a non-expert perspective, minimizing technical jargon or providing clear explanations for specialised terms to enhance readability.

We have tried to minimize the technical jargon. We have also added Supplementary Notes 1 and 2 to explain the image analysis methods in more detail.

9) List the exact software or open-source tools used for sequencing and correlation analysis to ensure reproducibility of the methods.

Software and open-source tools mentioned in materials and methods have been referenced with the versions used. A renv lockfile has been uploaded onto github for users to access packages and dependencies used in the analysis as well.

Reviewer #2 (Remarks to the Author):

Summary: In this manuscript, the authors measured features of texture and granularity on phase contrast images of in house patient derived GSC stem cell lines across time points spanning the range of phenotypes seen as cells grow to confluence. A principle component analysis revealed two principle components that explained a large proportion of the variance seen. PC2 was found to correlate with transcriptional state in the lines matching a neuronal developmental phenotype.

Overall comments: The authors have provided novel exploration of a label free spatial analysis framework and evidence that these features may correlate with transcriptional state within GSC lines. Evidence for this result would be significantly strengthened by testing the findings on a validation set, i.e. derive or purchase a new set of lines, collect phase images and predict developmental state via the minimal required measurement set (perhaps informational measure 1 or 2).

We thank the reviewer for this recommendation. In response, we have incorporated four additional GSC datasets—each with matched phase-contrast imaging and bulk gene-expression profiles—and performed the requested analyses to predict the neurodevelopmental–injury response (NIR) gradient using minimal image features. The results are shown in Supplementary Figure 7 and Figure 4C.

In response to reviewer 1's major point #1, we trained a Lasso regularized regression model to predict position on the NIR gradient based on image features, which achieved a model performance of $R^2=0.788$, $p\text{-value} = 1e-05$ (Figure 4B). We also trained a linear regression model using only the two image features with the strongest top and bottom PC2 loadings— Informational measure 1 and Granularity 13. The linear model trained using the training set of 15 samples with these two features however did not show improvement over the Lasso model trained on the same training set and reported in the previous paragraph ($R^2=0.368$, $p\text{-value} = 0.0165$) (Figure 4C).

We finally used the Lasso model to predict the NIR scores for four new samples, which we used as our validation cohort. While difficult to fit with only 4 samples, we found a positive correlation trend ($R^2=0.288$, $p\text{-value} = 0.464$) (Supplementary Figure 7).

Specific comments:

1. Figure 2C and 3B – How was the order of the GSC lines on the x-axis here determined?

Because PC2 of the image features was the component most closely associated with the neurodevelopmental phenotype, we use each sample's PC2 score, representing its position along the neurodevelopmental–injury response (NIR) gradient, to define the x-axis ordering. Sample order along this axis was determined in two steps: first, we averaged the PC2 scores of all images within each sample's confluency group, then we averaged those group-level means across all confluency groups. Thus, the x-axis in Figures 2C-D and 3B-D reflects a mean PC2 position of each sample. We have clarified this in the figure caption.

2. Figure S2A – Why does the covered area vary so much for many of these lines to the degree that they are often largely overlapping across the confluence levels? The derivation of the confluence levels needs further explanation. Why not have defined cut points of covered area leading to distinct bins of confluence?

Technical and biological factors both contribute to the observed variability (now shown in Supplementary Figure 3A). Even within the same plate (and thus the same time point), differences in cell density among technical replicates, for example caused by different amounts of cells introduced on the plate at the start of the experiment, can push wells into different confluency bins if one applies a rigid area-based cutoff. Initially, we partitioned images purely by cell-occupied area but this interfered with biological correlation discovery. Correcting for technical plate effects enabled us to identify genuine multicellular organizational phenotypes correlated with transcriptomics information.

3. Figure S2A - Axis labels are too small to read. GSC line – plot color should stay consistent across the plots.

We have corrected the axis label sizes as well as ensured color consistency across the plots.

4. Figure 3B - Why does the number of lines differ across the confluence points?

When we calculated confluency areas, certain samples fell outside their expected bins. For example, larger cell size or higher seeding density produced high confluency readings even at early time points, whereas other samples never reached full coverage, thus some wells are classified in different confluency groups compared to other wells from the same time point. These biological growth dynamics and technical factors explain the disparities observed across confluency bins. We have now clarified this in the manuscript methods section.

5. Figure 3B - It appears that the GSC line to color pairing differs across the confluence points, this should be corrected. Ideally, we should be able to follow the same GSC line down the column in the same position and color to track how the measure is changing across confluence.

We apologize for this oversight. We have corrected the color pairings for Figure 3B-D. Every GSC line has a single color that is consistent across all panels in Figure 3

6. Figure 3B – it appears from the dot density comparing down columns, that for many of the GSC lines they achieved the maximum degree of confluence and then remained there for some time. It may be informative to plot these points over time to see if these metrics change over time after confluence is reached.

Figure 3B shows feature values (y-axis) changing across the samples, ordered by neurodevelopmental–injury response gradient (x-axis). The data points do not represent confluence values. For example, a highly confluent image may have a low or high feature value (e.g. the inverse difference moment feature) and a low confluent image may also have a low or high feature value. We have clarified this in the manuscript.